# Moving through Motherhood: Involving the Public in Research to Inform Physical Activity Promotion throughout Pregnancy and Beyond

**DOI:** 10.3390/ijerph18094482

**Published:** 2021-04-23

**Authors:** Victoria E. Salmon, Lauren R. Rodgers, Peter Rouse, Oli Williams, Emma Cockcroft, Kate Boddy, Luana De Giorgio, Ciara Thomas, Charlie Foster, Rosie Davies, Kelly Morgan, Rachel Jarvie, Christina Weis, Richard M. Pulsford

**Affiliations:** 1College of Medicine and Health, St. Luke’s Campus, University of Exeter, Heavitree Road, Exeter EX1 2LU, UK; v.salmon@exeter.ac.uk (V.E.S.); l.r.rodgers@exeter.ac.uk (L.R.R.); E.J.Cockcroft@exeter.ac.uk (E.C.); k.boddy@exeter.ac.uk (K.B.); led215@exeter.ac.uk (L.D.G.); ct537@exeter.ac.uk (C.T.); rachel.jarvie@exmouthcollege.devon.sch.uk (R.J.); 2Department for Health, University of Bath, Claverton Down, Bath BA2 7AY, UK; pr222@bath.ac.uk; 3Florence Nightingale Faculty of Nursing, Midwifery & Palliative Care, King’s College London, 57 Waterloo Road, London SE1 8WA, UK; oli.williams@kcl.ac.uk; 4The Healthcare Improvement Studies Institute, Clifford Allbutt Building, Cambridge Biomedical Campus, University of Cambridge, Cambridge CB2 0AH, UK; 5National Institute for Health Research Applied Research Collaboration South West Peninsula (PenARC), South Cloisters, St. Luke’s Campus, Exeter EX1 2LU, UK; 6Centre for Exercise, Nutrition and Health Sciences, School for Policy Studies, University of Bristol, Bristol BS8 1TZ, UK; charlie.foster@bristol.ac.uk; 7National Institute for Health Research Applied Research Collaboration West of England (ARC West), University Hospitals Bristol NHS Foundation Trust, 9th Floor, Whitefriars, Lewins Mead, Bristol BS1 2NT, UK; rosemary3.davies@uwe.ac.uk; 8Centre for Development, Evaluation, Complexity and Implementation in Public Health Improvement (DECIPHer), School of Social Sciences, Cardiff University, Cardiff CF10 3BD, UK; morgank22@cardiff.ac.uk; 9Centre for Reproduction Research, De Montfort University, The Gateway, Leicester LE1 9BH, UK; christina.weis@dmu.ac.uk; 10College of Life and Environmental Sciences, St. Luke’s Campus, University of Exeter, Heavitree Road, Exeter EX1 2LU, UK

**Keywords:** pregnancy, postnatal, post-partum, healthcare professionals, physical activity, policy, patient and public involvement, co-design

## Abstract

Information received by women regarding physical activity during and after pregnancy often lacks clarity and may be conflicting and confusing. Without clear, engaging, accessible guidance centred on the experiences of pregnancy and parenting, the benefits of physical activity can be lost. We describe a collaborative process to inform the design of evidence-based, user-centred physical activity resources which reflect diverse experiences of pregnancy and early parenthood. Two iterative, collaborative phases involving patient and public involvement (PPI) workshops, a scoping survey (*n* = 553) and stakeholder events engaged women and maternity, policy and physical activity stakeholders to inform pilot resource development. These activities shaped understanding of challenges experienced by maternity and physical activity service providers, pregnant women and new mothers in relation to supporting physical activity. Working collaboratively with women and stakeholders, we co-designed pilot resources and identified important considerations for future resource development. Outcomes and lessons learned from this process will inform further work to support physical activity during pregnancy and beyond, but also wider health research where such collaborative approaches are important. We hope that drawing on our experiences and sharing outcomes from this work provide useful information for researchers, healthcare professionals, policy makers and those involved in supporting physical activity behaviour.

## 1. Introduction

Physical activity has positive benefits for maternal and foetal health during and after pregnancy, including reduced risk of excessive gestational weight gain, gestational diabetes mellitus and symptoms of postnatal depression [1,2,3,4,5,6]. Yet, during pregnancy, many do not meet physical activity levels recommended for this time. For example, data from the US suggest that only 32% of pregnant women meet public health guidance for weekly physical activity, and that this may be as low as 12% later in pregnancy. [7]. Pregnancy has been proposed as an opportune moment for promoting engagement in physical activity as a healthy behaviour [1,8]. However, information received regarding physical activity during and after pregnancy often lacks clarity, with women often receiving conflicting and confusing advice from their antenatal care providers [9,10,11]. In recognition of this, the UK Chief Medical Officers (CMOs) published physical activity guidelines in 2019 and developed infographics depicting evidence-based guidelines for physical activity in pregnancy [12,13] and the postnatal period [14].

While this policy initiative makes important steps towards summarising and operationalising the evidence base, it is necessary to remember that these infographics were designed to support healthcare professionals (HCPs) to promote physical activity rather than pregnant women or mothers to engage with them independently [13], though they are now commonly being used in this way, e.g., displayed in General Practice waiting rooms. Women may have limited contact with HCPs during pregnancy, potentially limiting the reach and utility of the evidence-based recommendations communicated by the infographics [15]. In addition, there is evidence to suggest that women may prefer physical activity advice that focuses on their individual needs and that fits within the context of their everyday lives, as opposed to generalised counselling that focusses only on risk and health outcomes, such as reduced risk of excessive weight gain and gestational diabetes mellitus [10,16,17].

Pregnant women and new parents often seek information from alternative sources outside of traditional healthcare services, such as social media, where pregnancy may be framed as a medical ailment, with the potential to negatively influence attitudes to physical activity [18]. In addition, the rise of so-called parenting expertise has increased social pressure on new parents. Recent generations are overwhelmed by choices and responsibilities and can be left feeling anxious about whether they are good enough parents [19]. Without clear, engaging and easily accessible guidance centred on the experiences of pregnant women, and new parents, the benefits of physical activity during/after pregnancy could be dismissed by some as unachievable, and seen by others as being one more thing to worry about. Vital to creating such guidance is the active involvement of, and collaboration with, a diversity of people with experience of pregnancy and parenting in the co-design of any resource which aims to communicate the benefits of and possibilities for physical activity during pregnancy.

In recent decades, the involvement of, and collaboration with, patients, members of the public and others with relevant lived experience (rather than professional expertise) in health research and healthcare service design has become increasingly commonplace [20,21]. However, this collaborative approach is not commonly adopted in the development of physical activity promotion information, resources and communication, [22,23]. Numerous rationales for more inclusive and collaborative ways of working are invoked and contested but can broadly be separated into two groups: democratic and technocratic. That is, it is important to involve those with relevant lived experience in health(care) research and service design because (i) such research and services tend to be publicly funded and citizens therefore have a democratic right to, at the very least a conduit to, influence decision-making and (ii) research and services can be made more fit for purpose if their design, delivery and evaluation are informed by the needs and preferences of those with relevant lived experience [24]. The Moving Through Motherhood project started from the premise that working in collaboration with pregnant women, mothers and other key maternity and physical activity stakeholders would make our research and service development both fairer and more fit for purpose than if a less inclusive and collaborative approach was adopted.

The Moving Through Motherhood collaborative process aimed to explore the physical activity experiences and needs of women during and after pregnancy. This paper uses the example of Moving Through Motherhood to demonstrate the value of this collaborative process in the design of evidence-based, user-centred physical activity information and resources which reflect diverse experiences of pregnancy and early parenthood and to argue for the normalisation of such approaches in this field.

## 2. Materials and Methods

The Moving Through Motherhood project consisted of two iterative phases which are described in Figure 1: Phase I: (a) patient and public involvement (PPI) workshop 1, (b) online survey, and (c) pilot resource development 1; Phase II: (a) stakeholder event, (b) PPI workshop 2, (c) and pilot resource development 2.

Ethical approval for the online survey was granted by the University of Exeter College of Life and Environmental Sciences Research Ethics Committee (Reference: 171206/A/03).

### 2.1. Phase I

#### 2.1.1. Phase Ia: PPI Workshop 1

In Phase I in October 2017, a PPI advisory group of women (*n* = 4) with children under 3 years old, who were members of an advisory group for other research relating to women’s health in and around pregnancy, reviewed existing UK guidelines and infographics for physical activity in pregnancy [12,13] and discussed the design of an online survey. Women were invited to join the group via their local children’s centre that was located in a socially deprived area of the city of Exeter, in South West England, and that serves a relatively diverse catchment. Demographic data were not formally collected as the women were public advisors, not research participants. The group met in a local play café and meetings were facilitated by the research team. Women were able to attend with their children to minimise childcare concerns. PPI advisors were given a thank you payment of £25 for attending each workshop.

#### 2.1.2. Phase Ib: Online Survey

An online survey containing open and closed questions was designed with the Phase I PPI group using the Online Survey platform (formerly Bristol Online Survey) (https://www.onlinesurveys.ac.uk/; accessed on 26 March 2021) (see Appendix A). The survey contained 27 questions relating to demographic information (9), physical activity views/experiences (11) and physical activity guidelines/information (7). The purpose of the questionnaire was to further explore expectant mothers and parents experiences of physical activity during and after pregnancy, and of any information or guidance they received at this time (including where it came from and how useful it was perceived to be). The PPI advisory group piloted the survey, and questions were refined in response to their feedback prior to distribution. The survey was distributed to a convenience sample via a wide range of online regional and national mothers’ groups and forums (with the aim of reaching their significant and diverse usership), research team collaborators and social media networks, and PPI group members’ personal networks. The survey was open from 27 November to 22 December 2017 and invited adult women over 18 years old with experience of pregnancy to take part. Quantitative survey data were imported into Stata v13 (StataCorp. 2013; College Station, TX, USA) for descriptive analyses. Free-text responses were analysed using thematic analysis to identify key themes [25].

Following data collection and analysis, the PPI group reviewed results and key themes arising from the survey, discussed what they perceived to be information gaps and highlighting key areas for further exploration.

#### 2.1.3. Phase Ic: Development of Pilot Resources 1

Key themes from the survey were discussed with the Phase I PPI advisory group. The group worked with members of the research team to create ‘like me’ stories based on participants responses to open questions. Advisors were presented with a summary of key themes and direct quotations from the survey. The group selected sample quotations and used these to create short vignettes using a three-stage format relating to (1) concerns and challenges regarding physical activity, (2) positive experiences and (3) helpful advice and tips for other pregnant women and new mothers. Advisors chose to create the selected stories based on common concerns identified in the survey and confirmed by the women’s own experiences. The stories were designed to be narrated by fictional characters. A graphic designer illustrated the characters and their stories and developed a pilot web page to host the pilot resources.

### 2.2. Phase II

#### 2.2.1. Phase IIa: Stakeholder Event

Phase I PPI advisors, additional PPI representatives and other key maternity stakeholders were invited to contribute to a regional stakeholder event in June 2019. Delegates (*n* = 50) at the event included academics (*n* = 11), clinical HCPs, (*n* = 20) public health practitioners and government/local authority representatives (*n* = 7), physical activity service providers (*n* = 5), and charity representatives (*n* = 6).

The objectives of the event were to understand the reality and challenges for supporting physical activity during and after pregnancy, identify potential solutions, and explore ideas for what is needed to put solutions into practice. Delegates also reviewed and commented on the suitability and usefulness of pilot resources from Phase I. The event consisted of presentations of Phase I findings, round table discussions with mixed stakeholder groups and small group discussions facilitated by the Moving Through Motherhood research team.

#### 2.2.2. Phase IIb: PPI Workshops 2

Pregnant women and women with young children attending children’s centres in two regional cities, Exeter and Bristol, in South West England, who had not been involved in earlier research, were invited to attend a series of two further PPI workshops in September and October 2019. In Exeter, the same group of advisors (*n* = 11) attended both workshops, whereas in Bristol two different groups of advisors attended the first (*n* = 8) and second (*n* = 5) workshops. Organisation of PPI meetings was facilitated by the National Institute for Health Research (NIHR) Applied Research Collaboration Southwest Peninsula (PenARC), ARC West and local children’s centres. We specifically liaised with children’s centres in areas of the cities serving diverse ethnic and socio-economic populations to seek broader involvement of women with different perspectives. The groups reviewed and provided feedback on findings from Phase I, including pilot resources, and the Phase II stakeholder event to inform the direction of future research and resource development. As in Phase I, women attended with their children, were given either a thank you payment or vouchers worth £25 for each workshop.

#### 2.2.3. Phase IIc: Development of Pilot Resources 2

In response to ideas and suggestions from the stakeholder event and PPI workshops 2, the research team worked with a film maker to produce pilot films to represent four of the example stories created by PPI advisors in Phase I.

## 3. Results

### 3.1. Phase I

#### 3.1.1. PPI Workshop 1

PPI advisors gave feedback on potential pitfalls that may occur when relaying information from the CMO infographic [12,13] to pregnant women. Although the infographic was designed for HCPs, the group noted that terminology, such as “moderate intensity activity”, “muscle strengthening”, “do not bump the bump” and “no evidence of harm”, were not clearly understood, and may not be conducive to promoting physical activity or reducing a sense of risk when used in public facing information. The workshop shaped the design of an online survey, with advisors proposing questions based on their discussions, suggesting a maximum time for completion of 10 min, recommending optimisation for completion on a smartphone, and piloting and refining the survey before final distribution.

#### 3.1.2. Quantitative Survey Findings

Survey respondents (*n* = 553) were predominantly located in England, UK (77.9%), aged between 35 and 44 years (49.9%), educated to at least undergraduate degree (83.2%), and of white ethnicity (91.1%) (Table 1). Respondents were mothers (79.2%), pregnant (11.8%) or both (8.5%). The majority had one or more children (88.6%), with 70.5% having a youngest child aged 2 years or less. The research team acknowledged that the survey respondents lack diversity and are relatively well educated and responded by prioritising targeted PPI activities with groups who were absent or under-represented in this sample.

An overwhelming majority of respondents (90%; *n* = 497/553) thought that it was ‘important’ or ‘very important’ to be physically active during pregnancy. However, despite the majority of respondents being ‘quite’ or ‘very active’ prior to pregnancy (71%; *n* = 394/553), and from social groups that typically have fewer barriers to and consequently are more physically active [26,27,28], most (60%; *n* = 334/553) were less active during pregnancy than before or not at all active during their pregnancy. Trimesters three (61%, *n* = 338/553 of respondents) and one (33%; *n* = 180/553) were viewed as the least active stages of pregnancy and two was considered to be the trimester most women felt able to be most physically active (64%; *n* = 353/553). Over half of respondents (56%; *n* = 310/553) felt ‘confident’ or ‘very confident’ about being active during pregnancy but this still left a substantial number who were ‘unsure’ (13%; *n* = 74/553) or ‘not very’ or ‘not at all’ confident (27%; *n* = 148/553). Most respondents were not aware of formal guidelines and recommendations for physical activity during pregnancy (67%; *n* = 373/553), although 76% (*n* = 420/553) did receive or read some information about physical activity during pregnancy and found it ‘quite’ or ‘very’ useful (65%; *n* = 271/420) [29].

#### 3.1.3. Qualitative Survey Themes

Free-text survey responses indicated that many women felt physical activity was important, believing keeping active was beneficial for general health and wellbeing for mother and baby, and to help with labour and recovery. Respondents also noted benefits for mental health and mood. Yet, confidence with participating in physical activity was undermined by a range of factors. Four key themes were identified from qualitative survey data. Themes related to the reality and challenges for physical activity during and after pregnancy, unmet needs and potential solutions:

**1. Communication of physical activity guidance:** Existing advice/information was felt to be vague, conflicting and confusing and often assumes that women are already physically active, which may alienate those who are not.

*Need/solution* = ‘Professional’ advice, e.g., from a health or fitness professionals, is a source of reassurance and encouragement for physical activity. More specific advice on suitable types and amount of physical activity would also be welcome.

**2. Safety concerns and fear of judgement hinder participation in physical activity:** Uncertainty about what physical activity is safe, fear of harm to mother and/or baby and fear of judgement by others (as being irresponsible or doing the wrong things) undermines confidence to engage in physical activity.

*Need/solution* = Information supporting physical activity should directly address safety concerns.

**3. Being active is not always easy!** Although highlighting benefits of physical activity for improving symptoms of pregnancy, (e.g., fatigue) and mental health can be helpful, pressure to be active may have a negative impact if someone is struggling to participate in physical activity due to pregnancy-related complications or social constraints.

*Need/solution* = To recognise the importance of being kind and gentle with yourself when pregnant, and for tailored information about physical activity which is specific to individual needs and circumstances, as well as advocating for equitable physical activity opportunities for those who are pregnant or parents—see below.

**4. Specific resources for pregnancy activities:** There was felt to be a lack of information and opportunities for physical activity, including accessible facilities and classes, tailored to the needs of pregnant women and parents.

*Need/solution* = Tailored resources might include tailored activities that offer more appealing, supportive and equitable opportunities to be active during and after pregnancy as well as tips and advice for fitting physical activity into daily life. Resources might also include information about—where they exist—local opportunities for pregnancy-related physical activity, for example, some councils offer free swimming for pregnant women.

A detailed summary of qualitative survey data, including illustrative quotations, can be found in Appendix A. These findings were confirmed and expanded by stakeholders and PPI advisors in Phase II.

#### 3.1.4. Phase I Pilot Resource Development

The phase I PPI group chose five initial ‘like me’ stories to develop, relating to, (i) safety concerns, (ii) nausea and fatigue, (iii) pelvic pain, (iv) timing and duration of physical activity in pregnancy, and (v) explaining recommendations for intensity of physical activity (for example, see Table 2). Fictional characters and their stories were illustrated by a graphic designer for two stories and housed on a pilot web platform (for example, see Figure 2. For other examples see Appendix A).

### 3.2. Phase II

#### 3.2.1. Summary of Phase II Stakeholder and PPI Workshops

Stakeholders and PPI advisors in Phase II were in broad agreement with Phase I themes, emphasising a need for clear, concise physical activity information that feels trustworthy, is evidence-based, considers key challenges such as pregnancy-related symptoms and safety concerns, and which acknowledges external roles and responsibilities, including childcare, work commitments and local opportunity (or lack thereof).

Delegates at the stakeholder workshop discussed a range of challenges and opportunities for improving physical activity advice and support in pregnancy and early motherhood, relating to healthcare solutions, social awareness, and resource requirements. Discussions mainly considered opportunities at a policy or organisational level. Healthcare-focused solutions explored ideas such as better integration of physical activity advice and support in antenatal care, increased access to physiotherapy and improved training and resources for HCPs—there was a strong sense that at present, it is a sidelined and largely unsupported issue. Stakeholders noted that solutions must consider ways to address social norms regarding physical activity during pregnancy and alleviate judgement and pressure from public and media. In addition, solutions should seek to promote better mental and emotional wellbeing from active pregnancy, and present family-centred solutions to support physical activity (for example, childcare solutions and whole-family physical activity activities) rather than focusing solely on reducing risk for physical health outcomes. The need for policy level support for physical activity, (for example, from national and local government, and HCP bodies), and adequate financial resourcing to enable implementation of potential equitable solutions was stressed. Stakeholders highlighted the potential benefits of population level media campaigns to shift social norms regarding physical activity in pregnancy, to include public, social media and television marketing. Campaign resources should include positive imagery on posters or advertisements, placed in gyms, medical or healthcare settings and public spaces, free ‘like me’ videos where women discuss their experiences, training resources for HCPs and trustworthy advice available for parents from one place (for example, a dedicated website) with a recognisable visual identity, such as a logo.

PPI groups in Phase II broadly agreed with stakeholder suggestions. Advisors reiterated the need for consistent messaging on physical activity from HCPs, with better education resources to support understanding of benefits of physical activity and address safety concerns. Information and resources need to be culturally accessible, allowing for non-native English speakers and cultural differences. Discussions highlighted the importance of achieving a sense of normality through engaging in physical activity, particularly acknowledging benefits for mental health. Advisors commented on a lack of facilities and classes available for pregnant women and acknowledged variation in social constraints and environmental opportunities for physical activity, for example, lack of time, money, access to suitable green spaces or safe places for walking.

The choice of language and definitions used to support communication about physical activity were crucial. Conversations in the stakeholder workshop tended to discuss physical activity in the context of formal structured exercise, such as free fitness classes, rather than less structured day-to-day activities such as walking. Although one PPI group in location B also discussed formal structured exercise, such as free swimming classes, the group in location A focused on more general physical activity, including walking for transport or as a more unstructured leisure time activity. These advisors returned for a second workshop and discussed how they had been thinking about the nature of physical activity between sessions and were either surprised by how much they had been doing or had purposefully done some extra walking to the shops in order to fit more physical activity into their day. For these women, understanding that physical activity does not have to mean formal and structured exercise enabled them to reframe their view of public health physical activity recommendations from something unachievable to something realistic and manageable within their daily lives; if someone was feeling isolated and stuck at home, a walk outside would ‘count’.

These discussions helped to shape our understanding of the current challenges experienced by both maternity and physical activity service providers, pregnant women and new mothers in relation to supporting physical activity, including how current information provision may not be sufficient to address individual concerns, social perceptions of physical activity in pregnancy may undermine confidence, and poor understanding of physical activity versus exercise may limit opportunities for engagement.

#### 3.2.2. Feedback on Phase I Pilot Resources and Recommendations for Future Development

Workshop discussions were invaluable for seeking feedback on the potential acceptability, utility and application of the pilot resources from those who deliver and receive physical activity information and support. The resources provided a starting point to stimulate discussion and explore ideas for future resource development.

Overall, stakeholders and PPI advisors responded positively to Phase I pilot resources. The ‘like me’ stories were perceived to be useful, relatable, reassuring, accessible, and informative. Delegates and advisors liked the format of problem and solution and felt that the scenarios appreciated the day-to-day variability that pregnancy presents. PPI advisors felt that the examples captured key challenges for engaging in physical activity, such as pregnancy-related symptoms, and fear of harm. The personal name tag, cartoon and age added a personal touch which made the stories feel real. Including a wider variety of women, for example of differing ages and ethnicities, will allow more women to relate to the scenarios and help increase their reach and success. Verbatim notes with suggestions for further development recorded at the stakeholder and PPI workshops are summarised in Table 3.

#### 3.2.3. Phase II Pilot Resource Development

Discussions with stakeholders and PPI advisors influenced our approach to the next stage of resource development. Following recommendations for video resources to tell the Phase I stories, the team collaborated with film maker to make four pilot films. The research team worked with a film maker to develop a script, and the stories were filmed using actors and actresses to bring the characters and their experiences to life. The films will be used as example resources in Phase III of Moving Through Motherhood, where the research team will continue to collaborate with women and stakeholders to co-design relevant, accessible resources to support physical activity during and after pregnancy.

## 4. Discussion

The Moving Through Motherhood project aims to support physical activity in pregnancy and beyond. This paper reports the processes undertaken to actively integrate key stakeholders and PPI into our research processes in a meaningful way, in order to strengthen the usability and suitability of research content, processes and outcomes. These processes may have application in other physical activity or public health research, or indeed research in other fields.

Women were involved in the Moving Through Motherhood project from the outset, designing survey questions, interpreting and commenting on findings, creating ‘like me’ stories based on survey responses, identifying the reality and challenges for supporting physical activity during and after pregnancy, identifying potential solutions, and exploring ideas for what is needed to put solutions into practice. Women and stakeholders also contributed to the design, refinement, suitability and usefulness of initial resources at multiple time points throughout the phases I and II of Moving Through Motherhood. Collaborating with women supported the research team to ask questions that were relevant and meaningful to pregnant women, and ensured that the research explored appropriate challenges, concerns and opportunities for improving physical activity guidance during and after pregnancy.

Overall, our findings confirm previous research that women who seek information about physical activity during pregnancy often find it conflicting, confusing, or absent [9,10,30,31]. The development of evidence-based guidelines by the UK CMO expert working group represented a significant stride forward [12,14]. However, the resultant information and infographics were initially designed as a tool for HCPs to support the advice they give to women. The recommendation that it “should be available widely on surgery information boards and websites, antenatal classes, on the NHS Choice website, and anywhere that pregnant women may meet” [6] (p. 460), risks its misuse as a stand-alone physical activity promotion tool directed at women, rather than for the purpose for which it was originally designed. Our PPI work highlights that women may not independently understand or positively respond to some of the advice on the infographic, and that it may add to the confusion, inadvertently discouraging women from being physically active when pregnant. Similarly, stakeholders recognised a need for simple resources to support provision of physical activity advice. This reflects previous research findings that midwives want simple, reliable resources to support communication about physical activity in the antenatal period [32]. There is therefore a clear need for public facing resources and material designed by and for women to support physical activity during and after pregnancy, and the findings presented here illustrate some important considerations for the development of such resources, and for future research in this area:

### 4.1. The Importance of Collaboration, and Recognition of Intrapersonal Differences in Experience and Context

Survey participants, PPI advisors and stakeholders identified many intrapersonal challenges to engaging in physical activity, including changes in physical capability and pregnancy-related symptoms such as fatigue, nausea and pain, but also experience interpersonal challenges, including caring roles and other responsibilities. These challenges have been well-documented [16,31,33,34]. However, few studies have explored the challenges of negotiating physical activity within the context of women’s daily lives and individualised experiences [16]. Moving Through Motherhood stakeholders and PPI advisors emphasised that physical activity information and advice must take into account the individual needs and experiences of different women. By taking the approach of involving and collaborating with women and stakeholders throughout, we were able to develop relevant, relatable pilot resources, informed by women’s experiences of pregnancy, parenting and physical activity. Our participatory approach has been validated by Nobles et al.’s research on physical activity messaging with diverse adult groups in Bristol [23].

Previous research has often focused on the benefits of physical activity for reducing the risk of negative health outcomes, for example, excessive gestational weight gain and gestational diabetes mellitus [1,4,6], and public health messages often direct women to engage in physical activity to prevent these outcomes [35]. However, our work with pregnant women and mothers highlighted that messages and advice about fitting physical activity into everyday contexts and recognising social constraints as well as personal anxieties may be more meaningful to women than health outcome orientated statements such as “helps to control weight gain” or prescribed activity such as doing “150 min of moderate intensity physical activity per week”. This reflects other research findings that some women may primarily value physical activity as enjoyable, sociable, and necessary ‘me-time’ for self-care and space away from other children or duties, rather than considering the physiological benefits or an obligation to stay healthy [16,17,35].

Discussions throughout the Moving Through Motherhood project shaped our awareness and understanding of the need for resources to also acknowledge women’s complex roles within a family unit, as a mother/partner, and associated daily routines, as well as social expectations and judgements of others. For example, many women who did engage in physical activity experienced judgement and stigmatisation by others [18,30,36], further undermining their confidence to be active and making it difficult to normalise and promote physical activity during pregnancy [16]. Resources must look beyond the intrapersonal and individual motivations and recognise how physical activity can be negotiated within these wider contexts.

### 4.2. Importance of Clear Relatable and Practical Advice

Initial pilot resources, developed in these first phases of Moving Through Motherhood, present stories based on women’s voices, chosen by women in PPI workshop 1. Mothers and pregnant women were able to relate to the characters, identifying with their struggles with physical activity during pregnancy and alleviating some of the loneliness and helplessness pregnant women may feel towards physical activity [37]. Including tips for other women within the stories goes beyond just giving information, and their basis in real experiences may have more value for women who may seek experiential knowledge over and above medical information [38]. The resources were also well received by stakeholders and PPI advisors in Phase II, suggesting they are useful and useable not only by women themselves, but also those who might implement physical activity advice and support during and after pregnancy. These discussions stimulated conversation and ideas for additional resources and influenced our decision to develop alternative formats for delivery, resulting in the production of pilot films to represent each character and their personal story. These findings confirm other research that noted a desire for physical activity messages to be relayed by relatable characters via multiple modes of delivery, including video [23].

### 4.3. Importance of the Credibility

In line with other similar involvement research, women and stakeholders confirmed that physical activity resources need to be credible and trustworthy [23]. For our project, credibility included content (for example, evidence-based messages), delivery (for example, by a trusted source such as a healthcare professional) and the location of information (for example, on reputable websites). Other authors have also reported a desire of women with gestational diabetes mellitus for physical activity information to be delivered by a HCP as a credible source [39]. However, it is important to consider that a range of physical activity messaging and services, including local opportunities for pregnancy specific physical activity, provided by alternative influential and trustworthy sources, such as respected community individuals and organisations, may be required to increase their accessibility and reach, particularly within different social and cultural contexts [23].

### 4.4. Strengths and limitations

By involving pregnant women and mothers at all stages of this research and collaborating with a wide range of important stakeholders including HCPs, public health practitioners and providers, we were able to explore experiences of physical activity, and the provision and receipt of physical activity information during pregnancy. The important insights described above can inform continued development of resources and materials to support physical activity in pregnancy by the Moving Through Motherhood team and others, but also have application for other aspects of health promotion and research.

Involving both service providers and end users in the design of resources to support physical activity during and after pregnancy will ensure that they meet their needs, in a format that is useful, grounded in experiences that are relatable and relevant, and focused on messages that are meaningful and realistic for women, ensuring that physical activity feels achievable whilst recognising social constraints and the need for more equitable service provision. A central strength of this work is the engagement with women and relevant stakeholders throughout, from the early stages of survey design and distribution, through to informing and then feeding back on pilot resources. This collaborative process involved stakeholders from all levels of the system from policy makers through HCPs and academic researchers, to providers and end users. The establishment of this stakeholder community can be leveraged to optimise the development of resources to support physical activity during pregnancy and beyond for maximum reach, utility and success. Increasingly, PPI in the development of healthcare research, policies and practice is not merely recommended but expected and even required [40,41,42]. Researchers and practitioners concerned with developing information and resources to support physical activity would have much to gain from normalising more inclusive and collaborative approaches to research and service/intervention design—as illustrated by the positive outcomes from the Moving Through Motherhood project and the example of Nobles et al. [23].

It is important to note, however, that project finding is limited by the lack of socio-demographic diversity among PPI groups and survey respondents. This lack of diversity was disappointing given significant research team efforts to engage with women across diverse backgrounds. In both locations where PPI workshops were held, women were invited to attend via community centres that served deprived areas of the cities in an attempt to hear and include the voices of women across socio-economic strata and different cultural and ethnic backgrounds. The limited uptake of our invitations to take part in the workshops by women from a range of ethnic and socio-economic backgrounds may reflect broader issues than simply lack of diversity involvement. Firstly, it is recognised that women and families with socially complex lives, including immigrants or those not able to speak English often face challenges accessing education, health and social services [43,44] including maternity services [45]. Secondly, whilst our attempts to recruit women through existing community groups might have enabled engagement of often under-represented, overlooked and service-resistant families, the role and positionality of children’s centre managers as community gatekeepers may have inadvertently hindered our ability to access and engage women from different ethnic and socially diverse populations [43,46]. Such formal gatekeepers who work for a community organisation but are not necessarily perceived as members of the target community may be less effective at aiding recruitment than informal gatekeepers who live within the community [43,46]. Given the need to promote physical activity at the population level, future work should endeavour to include greater diversity of women. Use of a community engagement framework for underrepresented, overlooked and service-resistant families, and investing time in identifying and building relationships with appropriate community gatekeepers in future research may help to overcome some of these challenges [43,47]. Although our advisors highlighted that existing guidance seemed to speak mainly to women who were already active, implying that some were not active prior to pregnancy, we did not record information about the physical activity levels of the women attending the PPI workshops. Therefore, it is possible that the ‘like me’ stories were informed largely by the perspective of women who were physically active before pregnancy, similarly to our survey sample. Future work would greatly benefit from incorporating a range of views and stories from women who were previously inactive and became active during pregnancy or remain inactive and are not interested in physical activity. In addition, future research would design a more systematic study with a stratified sampling plan to improve the diversity of survey respondents. In light of the current COVID-19 pandemic and the likelihood of virtual platforms or need to wear face masks in face-to-face meetings, new methods for implementing engagement strategies, particularly with these already hard to reach groups, will also need further consideration and development.

## 5. Conclusions

The Moving Through Motherhood project adopted a collaborative involvement approach, working with women and other key maternity and physical activity stakeholders to understand both the realities of physical activity during pregnancy, and the need for information and resource to support it. Our findings highlight the desire for clear and practical information that is considered by women to be credible and relatable. Working collaboratively with women and stakeholders who seek to benefit from research outputs allowed us to appreciate the importance of understanding and representing diverse experiences and contexts in any new information and resources that seek to communicate about physical activity. This project demonstrates that a collaborative approach is both achievable and important to support the iterative development of the research processes (methods) and outputs (resources). This example offers valuable insights that may inform further work to support physical activity during pregnancy and beyond, and also the wider fields of health research that can benefit from genuine involvement approaches.

## Figures and Tables

**Figure 1 ijerph-18-04482-f001:**
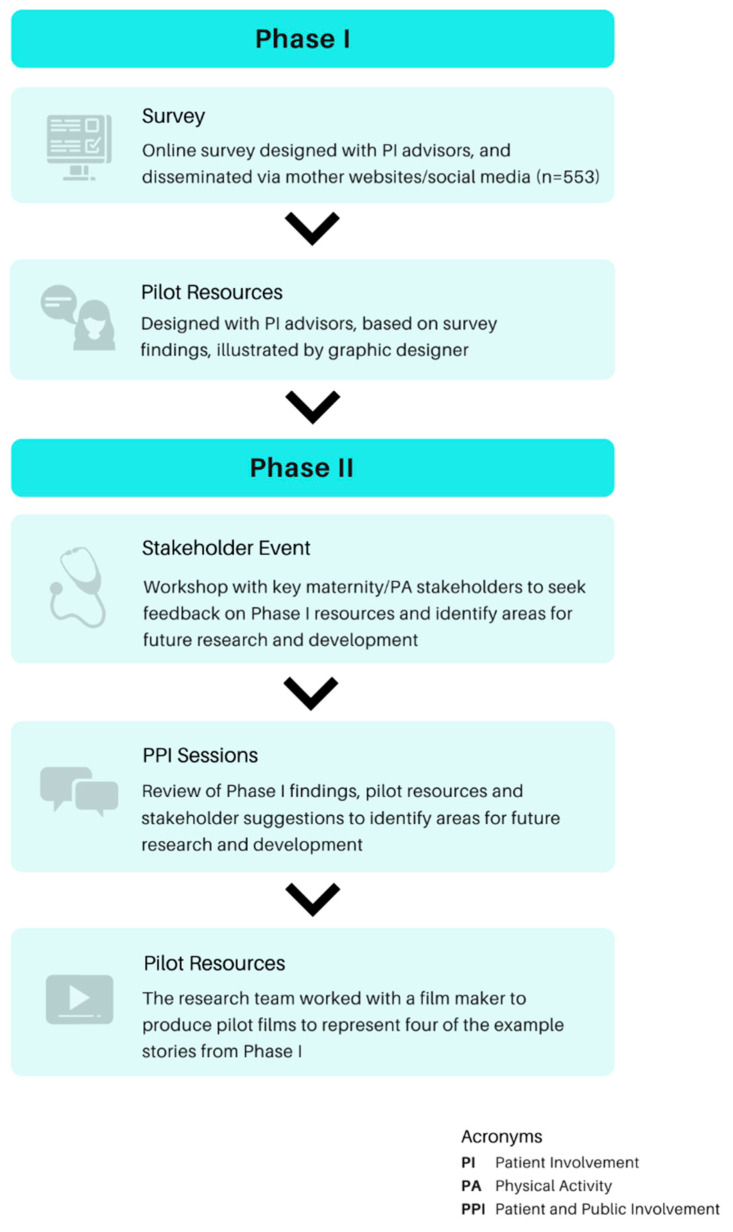
Phases of the Moving Through Motherhood project.

**Figure 2 ijerph-18-04482-f002:**
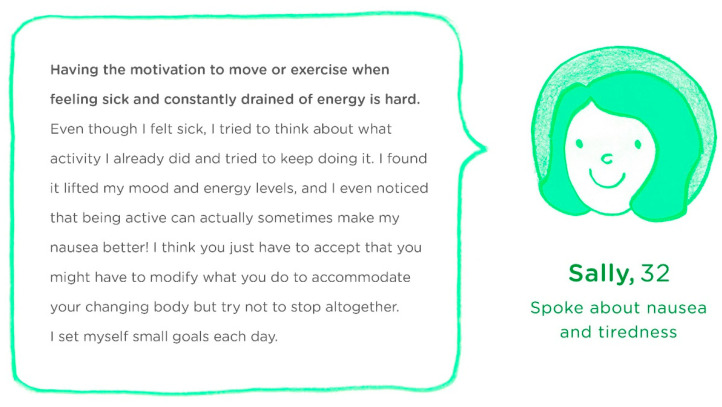
Example illustrated ‘like me’ story relating to nausea and fatigue.

**Table 1 ijerph-18-04482-t001:** Characteristics of survey respondents.

Characteristic	Category	Percentage (*n* = 553)
Age	18–24	1.8%
	25–34	44.9%
	35–44	49.9%
	45–54	2.0%
	55–64	1.3%
	Missing	0.2%
Location	Scotland	7.4%
	Wales	4.0%
	Northern Ireland	0.4%
	England	77.9%
	Other	10.3%
Education	Secondary	3.6%
	College/Further Education	10.9%
	Trade/technical/vocational	2.0%
	Undergraduate degree	32.0%
	Postgraduate degree	51.2%
	Missing	0.4%
Ethnicity	White	91.1%
	Mixed/multiple ethnic groups	3.6%
	Asian/Asian British	2.2%
	Black/African/Caribbean/Black British	1.5%
	Prefer not to say	0.9%
	Other ethnic group	0.5%
	Missing	0.2%
How many children	0	8.7%
	1	51.9%
	2	28.9%
	3	6.3%
	4	1.3%
	5	0.2%
	Missing	2.7%
Currently	Pregnant	11.8%
	A mother	79.2%
	Both	8.5%
	Neither	0.4%
	Missing	0.2%
Age of youngest child	0–2	70.5%
	3–5	11.8%
	6–10	5.2%
	11–15	1.5%
	16+	1.8%
	Not applicable	8.5%
	Missing	0.7%

**Table 2 ijerph-18-04482-t002:** Example ‘like me’ story template relating to nausea and fatigue.

Template Item	Selected Information and/or” Sample Quotation” from Survey Data
Background	Nausea, tiredness
Concerns	“Having the motivation to move/exercise when feeling sick and constantly drained of energy is hard.”
Positive experience/helpful advice	“Even though I felt sick I tried to think about what physical activity I already did and tried to keep doing it. I found it lifted my mood and energy levels, and I even noticed that exercise can actually sometimes make as nausea better!”
Tips for other women	“Accept that you might have to modify what you do to accommodate your changing body but try not to stop altogether. Set yourself small goals.”

**Table 3 ijerph-18-04482-t003:** Recommendations for further resource development.

Theme	Stakeholder/PPI Feedback
**Information delivery**	Create a video with real women or animation—use real-world experiences **Include healthcare professionals giving simple advice based on physical activity guidelines** *Write into soap script* *Include in leaflets* **Women do not want more leaflets** *Include “bigger” ladies in a video* *Use real voices*
**Content**	**Include a variety of ages** **Increase racial diversity** Make sure the information is reputable/trustworthy Use lay language for people with lower education/reading age (not the words moderate, nausea, modify, accommodate) Involve local maternity voices Cover more topics—myth busting Verbalise for different groups (dyslexic, visually impaired) **Include stories focusing on mental health, depression, anxiety—the emotional journey** **Include pelvic health** Use plain language *Do not like the word “exercise”* **Highlight difference between physical activity and exercise** Translate into different languages Mention cultural barriers and solutions *Use dialect* *At a glance not clearly about pregnancy*
**Format**	*Cartoons are childish, infantile, confusing, risk of patronising patients—use real images instead* **Cartoons, name and age give stories a personal touch** Visually a lot to digest—needs to be easy and simple Abbreviate messages to grab attention Make it less a block of text—paragraphs, different colours, more spacing, bold keywords *Make it look like a social media post—more familiar* *Add a link for further advice*
**Uses/dissemination**	**Disseminate information in early pregnancy, e.g., in first trimester** **Provide information in clinical settings** e.g., Film/animation and posters in waiting rooms, during clinic appointments, postnatal health visits **Make resources available in places frequented by pregnant women/new mothers, e.g., children’s centres, toddler groups, breastfeeding groups including their websites** **Make information available on trustworthy or recognised websites/platforms** **Have access to all information in one place, e.g., provide a single link to access extended resources** Put on Facebook pages, social media On websites—click + hear women talking *Can be used to start discussion in group workshops* *Instagram Live—expert answers questions* *Trip advisor of pregnancy* *Disseminate to Best Beginnings, Baby Buddy App*
**General comments**	Acknowledge that if you do not exercise, you do not fail as a mum Acknowledge that everyone’s ‘normal’ is different

PPI = patient and public involvement; *Italic text* = *ideas from stakeholders only*; **bold text = ideas from PPI workshops only**; all other text indicate ideas common to both groups.

## Data Availability

The data presented in this study are available on request from the corresponding author.

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
