# Peer review of "Moving through Motherhood: Involving the Public in Research to Inform Physical Activity Promotion throughout Pregnancy and Beyond"

_ijerph, 2021, doi:10.3390/ijerph18094482_

Round 1

Reviewer 1 Report

Authors adequately addresse all issues on the revised version of the article.

As I indicated previously, this is an interesting study aimed to explore the physical activity experiences and needs of women during and after pregnancy. The findings of this study have a great practical applicability and could improve how currently physical activity encouragement are faced. The process authors proposed may be a useful way to collaborate among all people involved on prenatal and post-natal care to encourage physical activity among pregnant women.

Author Response

Thank you for your comment. We are pleased that you feel we have adequately addressed the issues you raised in your review. Thank you for taking the time to review our manuscript.

Reviewer 2 Report

I thank the authors for their detailed and thoughtful responses to my earlier comments and am happy to recommend the manuscript for publication in its present form. 

Author Response

Thank you for your comment. We are pleased you are satisfied with our revisions. Thank you for taking the time to review our manuscript.

Reviewer 3 Report

Thank you to the authors for their thorough rebuttal and revisions. I have no further comments.

Author Response

Thank you for your comment. We are pleased that you are satisfied with our revisions

Reviewer 4 Report

Unfortunately, despite resubmitting the article, I still haven't received answers to most of my questions, which are listed below

  • What was the flow of participants? How many were sent the invitation to participate in the study, how many responded, how many completed the questionnaire, how many dropped out at subsequent stages of the study, etc.
  • What were the inclusion / exclusion criteria?
  • Did you calculate the minimum sample size?
  • What sample selection was used, was it a snowball method? Could the respondents send the questionnaire to other potential participants?
  • Did the persons fill in the informed consent to participate in the study?
  • Were factors such as SES, marital status analyzed (maybe there were some single mothers) 

Author Response

·         What was the flow of participants? How many were sent the invitation to participate in the study, how many responded, how many completed the questionnaire, how many dropped out at subsequent stages of the study, etc.

Thank you for your comment. This point was addressed in our response to the initial review but we are very happy to provide further clarity.

This manuscript describes a public and patient involvement and engagement process and as such the women and stakeholders involved in this work were not research participants. Therefore it is not appropriate to report their flow as research participants through the study.

We did not recruit directly for the initial scoping survey, which was undertaken to inform the scope of discussions with PPI advisors and stakeholders. The survey was shared via online forums (so did not get sent directly to individual women who were recruited in advance). As such we are unable to report response rate/attrition rates.

·         What were the inclusion / exclusion criteria?

This point was addressed in our response to the initial review and in the revised manuscript but we are very happy to provide further clarity:

For PPI workshops, we invited pregnant women and women with young children (most recent pregnancy <3 years) to attend PPI workshops. This is described in sections 2.1.1 and 2.2.2.

For the survey, we invited adult women with experience of pregnancy to complete the survey. We have added a sentence in section 2.1.2 to clarify this.

·         Did you calculate the minimum sample size?

This manuscript describes a public and patient involvement and engagement process and as such a minimum sample size was not required for any part of this work as described below.

Although we received a substantial number of survey responses, and in this manuscript we present some of the information obtained within the survey, this information is descriptive only. We do not (and have never intended to) make any statistical comparisons of effect, or undertaken any inferential analyses which would require a given level of statistical power.  

The objective of the scoping survey was to inform the focus of discussions with PPI advisors and stakeholders and to obtain qualitative data to inform development of resources. We aimed to get 500 responses to our survey. As this work was mostly qualitative in purpose this was seen as a manageable sample size for analysis. We used convenience sampling, a form of nonprobability sampling, as clarified in our recent response to reviewers. This type of sampling is suitable for pilot work, such as this. We used the information from the survey to inform the next stage of discussions on this topic and to highlight areas we would need to target in the future (who did not respond to the survey) in terms of demographics and for where to send our survey, e.g. targeted groups.  Future research would design a more systematic study with a stratified sampling plan. We have added a sentence to the limitations section to clarify this.

It is not appropriate to describe a minimum sample size for PPI and stakeholder discussions in the context of a patient involvement and engagement process, as women and other stakeholders were not considered research participants. 

·         What sample selection was used, was it a snowball method? Could the respondents send the questionnaire to other potential participants?

We added to our original response on line 137. We now state

‘..The survey was distributed to a convenience sample via a wide range of online regional and national mothers’ groups and forums.’

Respondents were able to pass on the link to the survey to other potential participants.

·         Did the persons fill in the informed consent to participate in the study?

This point was addressed in our original response and we are very happy to provide further clarity. Informed consent is not required for PPI as participants are advisors and not research participants. Ethical approval was obtained for the scoping survey, and as such (and in line with University of Exeter Research Ethics Committee guidance for online research) consent for their anonymised responses being stored (in line with GDPR and University of Exeter data management requirements), and for information provided in the survey to be used to inform our PPI work was provided implicitly by respondents reading the survey information and beginning the process of completing and submitting the survey online. 

·         Were factors such as SES, marital status analyzed (maybe there were some single mothers) 

Further to our original review response, we hope this provides additional clarity:

These factors were not were not assessed in our scoping survey.

Women in the PPI advisory groups were not classed as research participants. This was a public involvement and engagement role. The GRIPP2 reporting checklist (https://www.bmj.com/content/bmj/358/bmj.j3453.full.pdf) states that methods should provide “a description of patients, carers, and the public involved with the PPI activity in the study". It is not normal practice to collect and report on demographic data.  As such, we did not formally record demographic information. Therefore we do not have formal data about their ethnicity, SES, PA background.

This manuscript is a resubmission of an earlier submission. The following is a list of the peer review reports and author responses from that submission.

Round 1

Reviewer 1 Report

This is an interesting study aimed to explore the physical activity experiences and needs of women during and after pregnancy. The findings of this study have a great practical applicability and could improve how currently physical activity encouragement are faced. The process authors proposed may be a useful way to collaborate among all people involved on prenatal and post-natal care to encourage physical activity among pregnant women.

Several issues should be addressed previously accept the manuscript:

Introduction section, Line 50, please update the reference.

Methods section, please provide more information about the survey on methods section (Line 127). In addition to Supplementary files, in this section information about the number and typology of questions, and the topic addressed on the survey should be provided to improve the comprehension of this section.

Discussion section, Lines 420 to 424. This argument should be improved by findings that have shown that physical health during pregnancy could be improved by simply increase daily PA reallocating this time from sedentary behavior, in line with the literature message “every step counts”. This reference could support the argument: https://doi.org/10.1111/sms.13566

Conclusion section, This argument “professional’ advice, e.g. from a health or fitness professionals, is a source of reassurance and encouragement for physical activity” is an important finding from this study and therefore should be included on the conclusion section.

Reviewer 2 Report

I enjoyed reading this paper. It is well-written and informative, easily allowing the reader to follow the research process, and I believe it describes a useful and important initiative. The study is aimed at clarifying women’s understanding of, and experiences with, physical activity advice during pregnancy, and at developing more appropriate promotional materials in the UK than previously available. A strong feature of the study is the extensive collaboration with representatives from the target population throughout all the phases of the study. 

I do not have many comments to improve the paper but there are a few issues that I think could use some clarification.

  1. You provide little information regarding the PPI group in Phase 1A, other than that the women all had children below 4 years of age. Given their role in the development of the pilot material and survey, it would be useful to know a bit more about this group, such as ethnicity, SES, physical activity background. Similarly, there is limited information about the PPI group in Phase 2. Although you mention that, to compensate for the lack of (I presume ethnic) diversity and relative socioeconomic privilege in the survey sample, you endeavoured to involve absent or under-represented groups in the PPI activities, you only comment in your discussion that you were not very successful in this, but there is no actual information about the participants in the method section, other than the women being pregnant and/or mothers with young children.
  2. The reason I think some information about the women’s physical activity background would be useful is that I wonder about a possible bias of the advisors towards having been physically active before pregnancy. Certainly the survey has been biased in this way (with 71% being ‘quite’ or ‘very’ active prior to pregnancy, line 211; it is also likely that people with an interest in physical activity would have been more likely to respond to the survey in the first place). The perspective of already active women who desire useful information about continuing physical activity during pregnancy may thus have dominated the promotional material that was developed, particularly in the ‘like me’ stories, with less (or no?) attention to women who were inactive or not sufficiently active before pregnancy. Are there ‘like me’ stories that involve women who became active during pregnancy? As you note in lines 50-51, pregnancy might be an occasion to prompt inactive women to become active, but it is not clear to me to what extent this perspective was adopted in the material you developed.
  3. Among the stakeholder were physical activity service and maternity providers and you mention that challenges experienced by this population were discussed (line 33), and that potential solutions for supporting physical activity during and after pregnancy were explored (lines 162-163), however, I did not see any data that described these challenges and potential solutions. Combined with the information that women need/want information about local opportunities for pregnancy-appropriate physical activity offerings, I think it would have been valuable to address this issue in some detail. I appreciate that the focus of the paper is on developing promotional materials but I think this issue is at least worth mentioning in the discussion.

Reviewer 3 Report

The authors present their process to design evidence-based, user-centered physical activity resources for pregnant individuals. The research is clearly articulated and demonstrates feasible and focussed ideas for the much-needed translation of guidelines into practice.

I have a few comments

  1. Line 47 – The authors state ‘Physical activity has positive benefits for maternal and foetal health during and after pregnancy, including reduced risk of gestational weight gain, ….’ I believe this should state ‘reduced risk of excessive gestational weight gain.’ Gestational weight gain in itself is a necessary part of healthy pregnancy.
  2. Line 49 – The authors state that ‘many’ women do not meet PA levels – I would suggest it would be better to state a numerical value here. There are many empirical studies that has objectively measured prenatal PA levels.
  3. Line 52 – ‘information received regarding physical activity during and after pregnancy often lacks clarity, with women often receiving conflicting and confusing advice.’ I find this sentence ambiguous, information received from whom? I think just ‘from HCPs’ needs to be added for clarity or adjust the word ‘received’ to information available or similar.
  4. Line 68 – ‘.. health outcomes, such as weight loss..’ Similar to point 1, I do not think we should be using the term weight loss with regards to health outcomes / PA in pregnancy. Women shouldn’t be losing weight during pregnancy. Reduced risk of excessive gestational weight gain. Please check this nuance throughout the manuscript.
  5. Figure 1 – Acronyms need defining. Consider adding the n at each stage.
  6. Line 119 – The PPI advisory group: Does the use of this group of 4 women with involvement in other women’s health research bias the survey from the offset? How were these individuals chosen – I guess through professional association. I would be interested to see sociodemographic details for this group, but I would imagine they reflect the survey respondents e.g., highly educated. As the researchers did not manage to attract a more diverse cohort in later stages, involvement of a more diverse advisory group in Phase I could have helped facilitate this.
  7. Line 134 – State that the survey was distributed using convenience sampling.
  8. The authors address their lack of diversity in their survey, and I think this is the biggest limitation of the study. I think this research is very important, however, the authors have unfortunately missed the particular groups that are underserved regarding physical activity and other health advice during pregnancy. The benefits of increasing physical activity in these underserved groups could have larger impacts on pregnancy health, therefore far more should have been done to engage broader socioeconomic groups. This is not something that can be adjusted in the current study, but potentially the authors would consider repeating this work with a specific focus on underrepresented groups or trying to adjust this for later phases of the Moving through Motherhood project.
    1. For clarity in the abstract, I would suggest adding some detail regarding the sociodemographic status of respondents. It should be clear that these outcomes relate specifically to a mostly highly educated, white cohort.
  9. 5% of individuals had children over the age of 3. Was there a specific reason for involving women with children of older ages (i.e., were pregnant prior to the release of CMO guidelines?). I wonder how the influence of time and improved research over recent years would influence prenatal experiences (especially in those with children 16+ although I recognise this is a very small number of the total cohort).
  10. It is unclear as to where some of the data described in lines 209 to 217 is collected from. I did not identify questions in the supplementary material that specifically asked about trimesters and physical activity. Some further clarification is required either in the methods or in the supplementary material.
  11. The inclusion of Table 2 and Figure 2 is very useful. I think it would be useful to include other examples for the other four ‘like me’ stories in the supplement.

Reviewer 4 Report

Dear authors, I have a few questions regarding the evaluation of your manuscript:

  • How were women recruited?
  • What was the flow of participants? How many were sent the invitation to participate in the study, how many responded, how many completed the questionnaire, how many dropped out at subsequent stages of the study, etc.
  • What were the inclusion / exclusion criteria?
  • Did you calculate the minimum sample size?
  • What sample selection was used, was it a snowball method? Could the respondents send the questionnaire to other potential participants?
  • Did the persons fill in the informed consent to participate in the study?
  • Were factors such as SES, marital status analyzed (maybe there were some single mothers) 
  • Did you assesses any relationship between results of questionnaire and i.e. educational status, age or professional status?